# Kernel composition in sorghum landraces revealed via analyses of genotype-by-environment interactions

**Chalachew Endalamaw**[1,2*], **Dagmawit Tsegaye**[1], **Angeline van Biljon**[2],
**Liezel Herselman**[2], **Maryke Labuschagne**[2]

**1** Ethiopian Institute of Agricultural Research, Melkassa Agricultural Research Centre, Adama, Ethiopia,
**2** Department of Plant Sciences, University of the Free State, Bloemfontein, South Africa

\* chalachew23@gmail.com

## Abstract

Sorghum stands out among cereals due to its rich bioactive compound content and resilience to varying climates, addressing common issues such as protein, iron (Fe), and zinc (Zn) deficiencies in humans. This study aimed to determine the impact of the genotype, environment, and their interaction on the chemical and physical properties of sorghum grain across locations and seasons. A total of 361 sorghum landraces and four commercial checks were grown for two consecutive seasons from 2020 to 2021 at Melkassa (MK20 and MK21), Jimma (JM20 and JM21) and Miesso (MS20 and MS21). Using genotype main effects with genotype by environment interaction (GGE) ranking biplots, stable and high-performing genotypes were identified. MK21 emerged as an ideal environment for starch, while MS20 proved representative for protein content. For Fe content, environments MS21, MK20, and MK21 were representative, while MS20 and JM20 were discriminatory. MS21 was identified as the most representative for Zn content. These findings underline the diverse and specific performance of sorghum genotypes across various environmental conditions and traits. This study identified sorghum landraces with high and stable starch and protein content, as well as high and stable concentrations of Fe and Zn. Notably, genotypes like G358, G218, G221, G161, and G171 were noted for their high mean protein contents and stability. Genotypes such as G175, G248, G137, and G142, which demonstrated superior performance in Fe, and Zn content, are regarded as excellent candidates for further evaluation and incorporation into breeding programs, offering significant potential to enhance nutritional stability across diverse agroecological regions. Their consistent performance also highlights their potential to address micronutrient deficiencies, contributing to enhanced human nutrition and food security.

## Introduction

Sorghum (*Sorghum bicolor* L. Moench) is a vital cereal crop, particularly in semi-arid regions of Africa and Asia, where it plays a crucial role in food security and nutrition [1,2]. With its inherent drought tolerance and adaptability to marginal environments, sorghum is a staple for millions of people in these areas. Variations in both physical and chemical

**Data availability statement:** All relevant data are within the paper and its Supporting Information files.

**Funding:** The author(s) received no specific funding for this work.

**Competing interests:** The authors have declared that no competing interests exist.

components of the sorghum grain were observed due to the diverse set of sorghum genotypes and variations in growth conditions [3]. Recent research [4] showed that genetic biofortification effectively enhances sorghum's nutritional quality, with Indian landraces like PYPS 2 and PYPS 13 excelling in protein, digestibility, and micronutrient levels, offering a sustainable solution for improved nutrition and productivity. The physical attributes of the grain, including kernel size and hardness, displayed more variability as a result of the interaction between genotype and location [3]. Beyond providing caloric intake, sorghum is increasingly recognized for its potential to enhance the nutritional status of vulnerable populations due to its considerable variation in grain nutritional content, including protein, Fe, and Zn concentrations [5,6]. These micronutrients are essential for addressing widespread deficiencies that contribute to malnutrition and related health issues, especially in developing countries.

Despite the importance of sorghum in contributing to human nutrition, the expression of nutritional traits is highly influenced by genotype-by-environment interactions (GEI), making the identification of stable and high-performing genotypes across different environments a challenge. Previous studies have demonstrated that improvement of micronutrient levels, such as Fe and Zn, in sorghum through breeding is feasible, with multi-location and multi-season evaluations essential for identifying stable, high-performing genotypes for grain nutrient content [7,8]. Understanding these interactions is critical for breeding programs that aim to improve sorghum nutritional quality and ensure consistent performance across diverse agroecologies. The genotype main effects and genotype-by-environment interaction (GGE) biplot analysis has become a powerful tool for analyzing multi-environment trial (MET) data [9,10,11]. By integrating both genotype and GEI effects, the GGE biplot facilitates the identification of genotypes that are not only high-performing but also stable across different environments [12,13]. Furthermore, this method allows for the characterization of testing environments in terms of their representativeness and discriminatory power, aiding in the selection of optimal testing sites for future breeding efforts.

Adaptability and yield stability are important measures for crop cultivation in different agro-climatic regions. The stability and adaptability of genotypes across different environments can be assessed using various statistical tools such as joint regression [14] and genotype main effects in addition to GGE biplots [15]. GGE biplots serve as the most efficient and commonly used multivariate models for determining genotype stability, adaptability, and ranking, as well as for identifying suitable mega-environments [9,10,16–18].

Most sorghum landrace studies have focused on the evaluation of grain yield, with less emphasis placed on grain quality traits using MET data. The environmental effects on grain quality on diverse sets of sorghum genotypes have not been extensively studied. Previous research [19] assessed the grain quality traits of Ethiopian sorghum accessions across various environments, highlighting substantial genetic diversity for selection within individual and combined environments. Although GEI has been used to assess the stability of improved varieties of sorghum using MET data [20,21,22], information on the stability of sorghum landraces for quality traits using GGE biplot models is lacking.

The extensive variety of sorghum cultivation areas, coupled with genetic diversity, presents considerable difficulties in ensuring consistent quality across different growing regions. Consequently, this research aimed to assess how diverse genotypes and various growing locations impact the chemical and physical properties of sorghum grain. The objectives include identifying genotypes that consistently maintain quality across diverse growing locations and examining the environmental interaction with sorghum quality attributes.

## Materials and methods

### Study areas and crop management

This study quantified GEI for the wet intermediate (Jimma) and dry lowlands (Melkassa and Miesso) agroecology from 2020 to 2021. The two dry lowland experimental sites are found in the central rift valley of Ethiopia. Jimma is in the wet intermediate agroecology with good rainfall in the cropping season. Both Miesso and Melkassa are lowland sorghum production areas in Ethiopia [23,24]. Tables 1, 2 and 3 present data on planting and harvesting dates, rainfall patterns, relative humidity, maximum and minimum temperatures, and soil analysis for macronutrients and organic matter across the three sites. Additionally, Table in S1 Table provides supplementary data related to these parameters. Seeds were drilled in the row at 75 cm between rows soon after plowing and then thinned to maintain 15 cm intra-row spacing at all sites. Basal fertilizer was applied at the recommended rate and other agronomic practices were applied for normal sorghum production in Ethiopia.

### Design and data analysis

The 361 Ethiopian sorghum landraces and four improved varieties were evaluated for genetic diversity and GEI in two consecutive seasons (2020–2021). A germination test was done on seeds in 2020 before planting in 2021. In 2020 and 2021, 354 sorghum accessions and four checks were evaluated, with seven sorghum accessions replaced in 2021, resulting in a total of 361 sorghum accessions and four improved check varieties. The only difference between trials

**Table 1. Long-term average temperature and rainfall properties for each trial site.**

| Location | Longitude | Latitude | Altitude (m.a.s.l) | Soil type | Rainfall (mm) | Minimum T°C | Maximum T°C |
|---|---|---|---|---|---|---|---|
| Jimma | 36°47′ E | 7°40′ N | 1 753 | Clay | 1 500 | 20 | 29 |
| Melkassa | 39°9′ E | 14°6′ N | 1 550 | Clay loam | 763 | 20.4 | 34 |
| Miesso | 39°21′ E | 8°30′ N | 1 470 | Clay loam | 570 | 16 | 31 |

m.a.s.l., meter above sea level; T°C, temperature in degrees centigrade.

**Table 2. The planting and harvesting dates of trials in two consecutive seasons.**

| Location | Planting date | | Harvesting date | |
|---|---|---|---|---|
| | Year 1 | Year 2 | Year 1 | Year 2 |
| Jimma | May 26, 2020 | May 6, 2021 | Dec 8, 2020 to Jan 8, 2021 | Dec 4–30, 2021 |
| Melkassa | Jul 15, 2020 | Aug 3, 2021 | Jan 1–20, 2021 | Jan 10–12, 2022 |
| Miesso | Jul 20, 2020 | May 11, 2021 | Nov 27- Dec 5, 2020 | Aug 30–Dec 24, 2021 |

**Table 3. Soil analytical data for some macronutrients and organic matter for the three trial sites.**

| Environment | Soil depth (cm) | pH | EC (µS/cm) | Textural class | %TOC | %OM | %TN | Av.P (ppm) |
|---|---|---|---|---|---|---|---|---|
| Jimma | 0–30 | 6.67 | 96.50 | Clay | 1.57 | 2.71 | 0.16 | 8.45 |
| | 30–60 | 6.88 | 72.70 | Clay | 1.21 | 2.09 | 0.13 | 8.33 |
| Melkassa | 0–30 | 6.79 | 190.80 | Clay loam | 1.29 | 2.23 | 0.08 | 7.95 |
| | 30–60 | 7.28 | 230.00 | Clay | 1.25 | 2.16 | 0.08 | 7.73 |
| Miesso | 0–30 | 7.25 | 139.70 | Clay loam | 1.06 | 1.83 | 0.09 | 8.73 |
| | 30–60 | 7.28 | 231.50 | Clay | 1.03 | 1.78 | 0.09 | 8.61 |

EC, electrical conductivity; TOC, total organic carbon; OM, organic matter; TN, Total nitrogen; Av.P, available phosphorus.

of the two seasons was the replacement of seven sorghum landraces that did not germinate in 2020 by seven different landraces in 2021.

Due to the large number of sorghum landraces included in this study, a partially replicated (p-rep) design was used in a row-column arrangement in across environments. Partial replication was applied in all trial designs, as confirmed by previous research investigations [25]. A multi-environment trial design with 354 landraces and four commercial checks, totalling 358 sorghum entries across three sites (Jimma, Melkassa, and Miesso) was used. Of the total of 354 sorghum landraces, a third (118) were replicated at each site and the replicated set was different at each site with a unique set of replicates per location (Table 4). The four checks were replicated at all sites. Each plot consisted of a single row, 4 m in length. The experimental design consisted of 10 columns and 48 rows, with each row containing a different sorghum landrace. A total of 48 sorghum landraces were organized per single column. This experimental design allowed for the testing of the effects of both genotype and environment, as well as GEI. The Burtless test was conducted within individual environments before the combined analysis across environments. The sorghum grain quality data collected from the six environments were subjected to a combined analysis of variance (ANOVA) using R software [26,27]. Each model consisted of a fixed effect for the targeted trait at each trial, random effects for genotypes within the trial, and spatial error for each trial [28]. The GGE biplot analyses were conducted using GenStat software, version 15 [29].

## Analysis of sorghum grain quality

A near-infrared spectroscopy (NIRS) calibration model was established at the Melkassa Agricultural Research Centre to assess sorghum grain quality traits, providing a rapid and cost-effective alternative to traditional, labour-intensive wet chemistry methods. The development of NIRS models for sorghum grain quality analysis involved utilizing over 108 Ethiopian sorghum accessions [30].

It was reported [30] that the NIRS calibration equations demonstrated high coefficients of determination for calibrations ($R^2c$ = 0.815) and slightly lower coefficients for cross-validations ($R^2cv$ = 0.589) for tannin content. Similarly, the total starch content showed high $R^2c$ = 0.983 with slightly lower $R^2cv$ = 0.910, while amylose content exhibited high $R^2c$ = 0.762 and slightly lower $R^2cv$ = 0.697. The standard error of calibration (SEC) and standard error of cross-validation (SECV) were minimal (SECV/mean = 3.8%, 1.2%, and 4.3% for tannin, total starch, and amylose, respectively).

In this investigation, sorghum landrace collection samples underwent NIRS scanning at Melkassa Agricultural Research Centre. The percentage of sorghum grain protein, ash, starch, amylose, amylopectin, and moisture, as well as the parts per million (ppm) of Zn and Fe, were determined using NIRS. An Inframatic 9500 NIR System Grain Analyzer was used to scan 200 g of cleaned whole grain for every landrace accession. The experimental design for NIRS

**Table 4. Multi-environment trial design of 354 sorghum landraces and four checks was replicated at all three sites.**

| Replicated landraces | Jimma | Melkassa | Miesso |
|---|---|---|---|
| 1–118 | 2 | 1 | 1 |
| 119–236 | 1 | 2 | 1 |
| 237–354 | 1 | 1 | 2 |

1, unreplicated; 2, replicated

analysis mirrored the field layout used at each site, ensuring consistency in the evaluation of grain quality traits.

### Broad-sense heritability

Broad sense heritability ($H^2$) was estimated from individual and combined ANOVA [31]. $H^2$ is classified as follows: 0–30% is considered low, 30–60% is considered moderate and >60% is considered high [32,33].

$$\text{Heritability}(H^2) = \frac{\sigma^2 g}{\sigma^2 p} \times 100$$

$$\sigma^2 p = \sigma^2 g + \sigma^2 e$$

Where: $H^2$ = heritability in a broad sense, $\sigma^2 p$ = Phenotypic variance, $\sigma^2 g$ = Genotypic variance, and $\sigma^2 e$ = Environmental variance.

## Results

### Genotypic variation and genotype by environment interaction

The combined ANOVA showed significant effects due to genotype, environment, and GEI for all traits studied ($P < 0.01$) (Table 5). High variability was evident in the sorghum genotypes for grain quality traits. Highly significant genotype ($\sigma^2 g$), environment ($\sigma^2 en$) and genotype x environment ($\sigma^2 gen$) interaction variance effects were present for all traits (Table 6). Amylose and amylopectin showed the highest $H^2$ of 75.6%, followed by Fe with a $H^2$ of 72.4%, indicating that genetic factors played a large role in these grain quality traits (Table 6). The ANOVA revealed that both genotype and environmental factors significantly influence sorghum grain quality, emphasizing the importance of considering these components in the evaluation of sorghums for potential end-use.

The starch, amylose, amylopectin, protein, Fe, Zn, ash, and moisture contents varied significantly across all environments (Table 6). The starch content ranged from 64.44 to 72.71% in genotypes G207 (ETSL101056) in MS20 and G4 (Bonsa) in MK21, respectively. The amylose content ranged from 18.72 to 20.09% in genotypes G294 (ETSL101486) in MS21 and

**Table 5. Combined analysis of variance for grain quality traits of 365 sorghum landraces across six environments.**

| Source | DF | Starch | | Protein | | Amylose | | Amylopectin | |
|---|---|---|---|---|---|---|---|---|---|
| | | SS | MS | SS | MS | SS | MS | SS | MS |
| GEN | 364 | 248.46 | 0.68** | 818.73 | 2.25** | 76.12 | 0.21** | 76.13 | 0.21** |
| ENV | 5 | 8456.11 | 1691.22** | 4090.01 | 818.00** | 108.24 | 21.65** | 108.22 | 21.65** |
| GEI | 1820 | 76.51 | 0.04** | 126.60 | 0.07** | 5.71 | 0.003** | 5.71 | 0.003** |
| Source | DF | Fe | | Zn | | Ash | | Moisture | |
| | | SS | MS | SS | MS | SS | MS | SS | MS |
| GEN | 364 | 269692.36 | 740.91** | 33710.67 | 92.612** | 168.75 | 0.45** | 133.96 | 0.37** |
| ENV | 5 | 51348.76 | 10269.75** | 43928.97 | 8785.79** | 136.07 | 27.21** | 3161.23 | 632.25** |
| GEI | 1820 | 28810.41 | 15.83** | 21092.66 | 11.59** | 54.68 | 0.03** | 186.44 | 0.10** |

**= Significant at 0.01 significance level;

* = Significant at 0.05 significance level; DF = Degrees of freedom; GEN = Genotype; ENV = Environment; GEI = Genotype by environment interaction; SS = Sum of squares; MS = Mean squares.

**Table 6. Summary of variance components standard errors, and heritability for quality traits.**

| Trait | Range | Mean | $\sigma^2$g | SE | $\sigma^2$gen | SE | $\sigma^2$en | H² |
|---|---|---|---|---|---|---|---|---|
| Starch (%) | 64.44 -72.71 | 68.75 | 0.30 | 0.06 | 0.29 | 0.11 | 2182.00 | 34.65 |
| Amylose (%) | 18.72-20.09 | 19.60 | 0.01 | 0.00 | 0.01 | 0.00 | 1967.00 | 75.58 |
| Amylopectin (%) | 78.91-81.28 | 80.40 | 0.01 | 0.00 | 0.01 | 0.00 | 1967.00 | 75.58 |
| Protein (%) | 5.35-13.33 | 9.55 | 0.45 | 0.05 | 0.227 | 0.06 | 1321.00 | 57.60 |
| Fe (ppm) | 3.75-93.13 | 32.28 | 11984.76 | 126.11 | 0.11 | 0.32 | 748.21 | 72.42 |
| Zn (ppm) | 19.31-95.68 | 28.06 | 4.38 | 0.67 | 4.32 | 1.79 | 250.70 | 58.20 |
| Ash (%) | 0.76-3.82 | 1.81 | 0.09 | 0.01 | 0.05 | 0.01 | 537.60 | 59.62 |
| Moisture (%) | 11.41-18.47 | 15.50 | 0.12 | 0.03 | 0.16 | 0.05 | 2933.00 | 32.28 |

$\sigma^2$g, genotype variance; $\sigma^2$en, environment variances; $\sigma^2$gen, genotype × environment interaction variances; SE, standard errors; H², heritability.

G278 (ETSL101410) in JM20, respectively. The amylopectin content varied from 78.91 (G278) in JM20 to 81.28% (G294) in MS21. Genotype G294 (ETSL101486) could be considered hetero-waxy sorghum due to its lower amylose and higher amylopectin values. The genotype played a major role in influencing the amylose content of the starch, with the majority of cultivated sorghums characterized by "normal" type starches containing both amylose and amylopectin. The protein content ranged from 5.35 to 13.33% in genotypes G142 (ETSL100735) in MK21 and G358 (IS38353) in MS20, respectively. The Fe content ranged from 3.75 to 93.13 ppm in genotypes G6 (DS10) in MK20 and G142 (ETSL100735) in MK20, respectively. The Zn content ranged from 19.31 to 95.68 ppm in genotypes G296 (ETSL101492) in MS21 and G264 (ETSL101325) in JM20, respectively. The ash content ranged from 0.76 to 3.82% in genotypes G78 (ETSL100365) in MS20 and G148 (ETSL100768) in JM20, respectively. Moisture content ranged from 11.41 to 18.47% in genotypes G278 (ETSL101410) in JM20 and G184 (ETSL100921) in MS20, respectively.

## Genotype main effects with genotype by environment interaction biplot analysis

**"Which-won-where" polygon of GGE biplot.** The GGE biplot polygon view identified the highest-performing genotypes by showing interaction patterns among genotypes and environments (Figs 1 and 2). Genotypes located at the polygon's vertices showed the best performance or worst in one or more environments. The starch vertex genotypes were G265, G84, G246, G4, G40, G110, G36, G264 and G355. The protein vertices genotypes were G358, G161, G232, G346, G346, G202, G142, G94, G207 and G98. G142, G137, G194, G311, G272, G6, G18, G325, G190 and G119 were selected for the Fe vertex. The Zn vertices were G264, G249, G296, G285, G245 and G135 (Fig 1). As a result, these four groups of genotypes were the most responsive to environmental interactions for starch, protein, Fe, and Zn, in that order.

In the "which-won-where" GGE biplot, lines from the origin divide the biplot into different sectors and create different mega environments (MGEs). In this study, all traits had two MGEs formed, except for Fe which only had one. For starch, environments JM20, MK20, MK21, MS20, and MS21 formed a joint MGE, whereas environment JM21 was a separate MGE for these traits. For protein, environments JM20, MK20, MK21, MS20, and JMS21 formed an MGE together, whereas environment MS21 was a separate MGE for this trait (Fig 1). For Zn, environments JM20, MK20, MK21, JM21, and MS21 formed a joint MGE, whereas environment MS20 formed a separate MGE for this trait, whereas for Fe, all environments formed a single MGE (Fig 2).

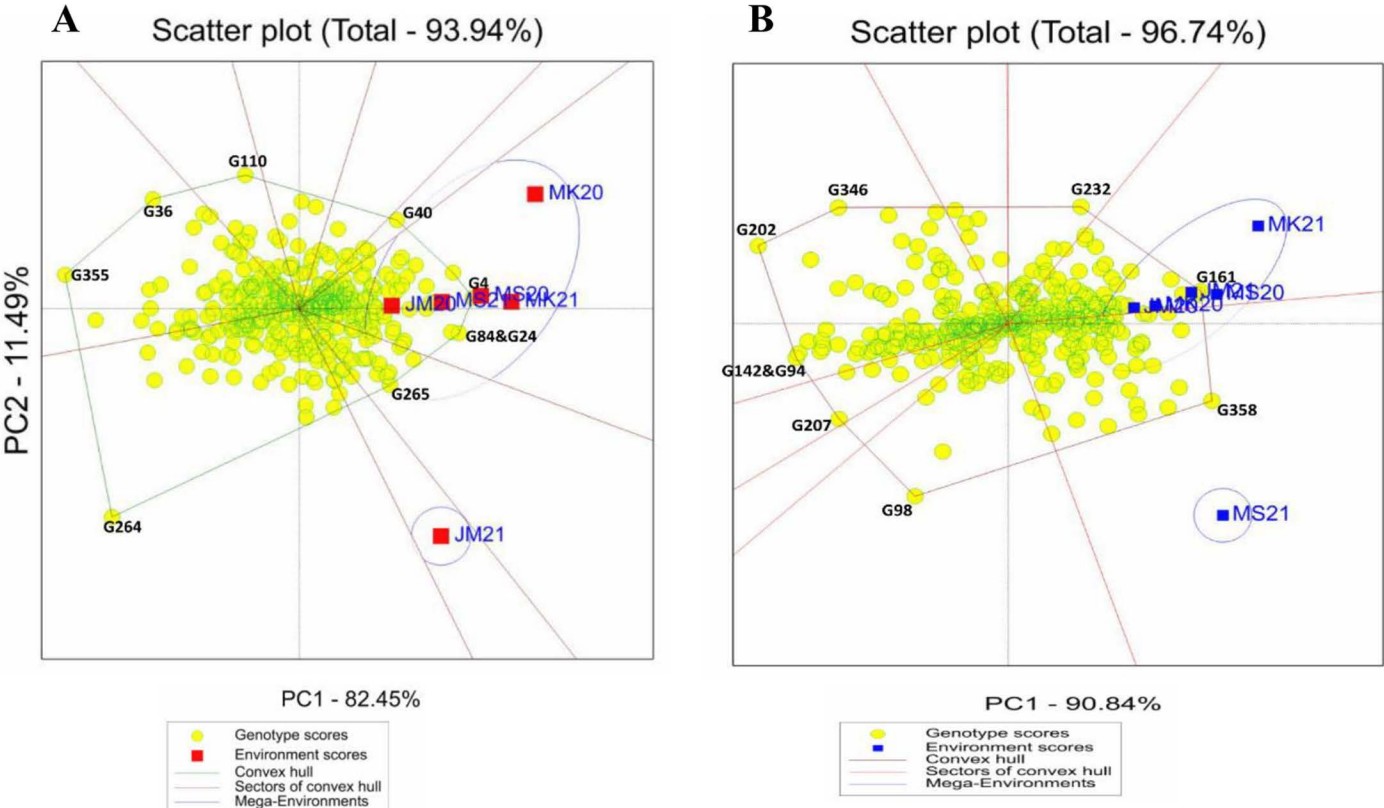

**Fig 1. "Which-won-where" polygon view of genotype main effects with genotype by environment interaction scatter biplot of the 365 sorghum landraces for (A) starch and (B) protein, displaying landraces with the best performance in each environment and the mega environments.** Vertex landraces on convex hulls (polygon) are the greatest for each mega environment for the correlating parameters. JM20 and JM21: Jimma 2020 and 2021, MK20 and MK21: Melkassa 2020 and 2021, MS20 and MS21: Miesso 2020 and 2021, respectively.

Genotypes G4 (Bonsa), G246 (ETSL101256), and G84 (ETSL100406) performed well in all environments for starch concentration. Genotype G161 (ETSL100834) performed well in all environments for protein concentration in 2020 and 2021, G358 (IS38353) performed well in MS21. Genotypes G355, G36, and G264 had low starch content, while genotypes G142, G94, G207, G202, and G98 had low protein contents in all environments (Fig 1). Genotype G264 (ETSL101325) performed well across environments for Zn concentration, whereas G245 (ETSL101255) and G135 (ETSL100798) performed well under the MS20. For Fe concentration, genotypes G137 (ETSL100702) and G142 (ETSL100735) performed well in across environments. Genotypes G272, G6, G18, and G325 had low Fe contents, while genotypes G285 and G296 had low Zn contents in all environments (Fig 2).

**Genotype performance and stability.** Genotypes should be assessed for average performance and stability over all environments within a single mega-environment. The single-arrowed line on a GGE biplot indicates the average environment coordinate (AEC) and indicates that the mean trait value is higher across environments. A genotype with a higher mean performance has an average environment axis (AEA) in the ranking biplot, represented by a single-arrowhead line passing through the origin. Thus, G246 and G84 had the highest mean starch content, followed by G4, G175, and so on; while G355 and G212 had the lowest mean starch content. In terms of protein, G358, G218, G161, G171, and G221 had the highest mean protein content, followed by G43, G88, G293, G184, and so on; while G202,

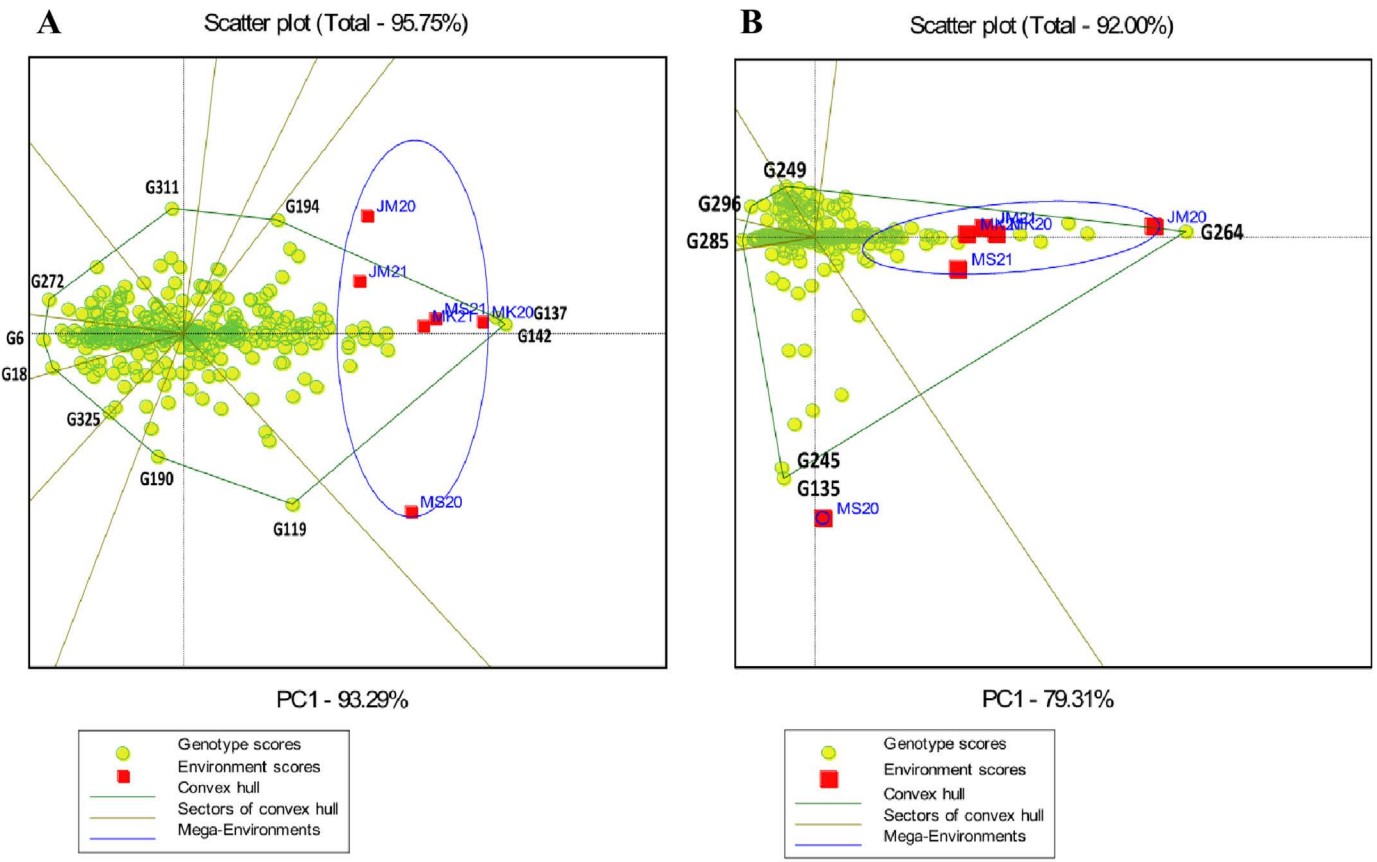

**Fig 2. "Which-won-where" polygon view of genotype main effects with genotype by environment interaction scatter biplot of the 365 sorghum landraces for** (A) Fe and (B) Zn, **displaying landraces with the best performance in each environment and the mega environments.** Vertex landraces on convex hulls (polygon) are the greatest for each mega environment for the correlating parameters. JM20 and JM21: Jimma 2020 and 2021, MK20 and MK21: Melkassa 2020 and 2021, MS20 and MS21: Miesso 2020 and 2021, respectively.

G63, G142 and G94, G346, and G207 had the lowest mean protein content (Fig 3). For Fe concentrations, G137 and G142 had the highest mean Fe content, followed by G166, G248, and so on. G272, G6, and G18 had the lowest mean Fe content. G264 had the highest mean Zn content, followed by G142, G278, G175, G148, and so on. G296, G289, and G111 had the lowest mean Zn protein content (Fig 4).

The stability of genotypes was assessed using the distance of the vector between genotype positions and the AEA in the ranking biplot. The best-performing and most stable genotypes are those that are far from the origin but on or near the AEA. As a result, G246 and G84 were the most stable genotypes with high mean starch content and the shortest vector from AEA, whereas G264 and G110 were the least stable genotypes with the longest vector from AEA. The highest mean protein content and shortest AEA vectors were found in G218, while the lowest mean protein content and longest AEA vectors were found in G232, G98, G346, G207, and G251 (Fig 3).

The highest mean Fe content and shortest AEA vectors were found in G137 and G142 and these genotypes were thus the most stable genotypes with high mean Fe content and shorter AEA vectors, whereas G311, G194, and G119 were the least stable genotypes with the longest AEA vector. Meanwhile, G148 and G175 had the highest mean Zn content and the shortest

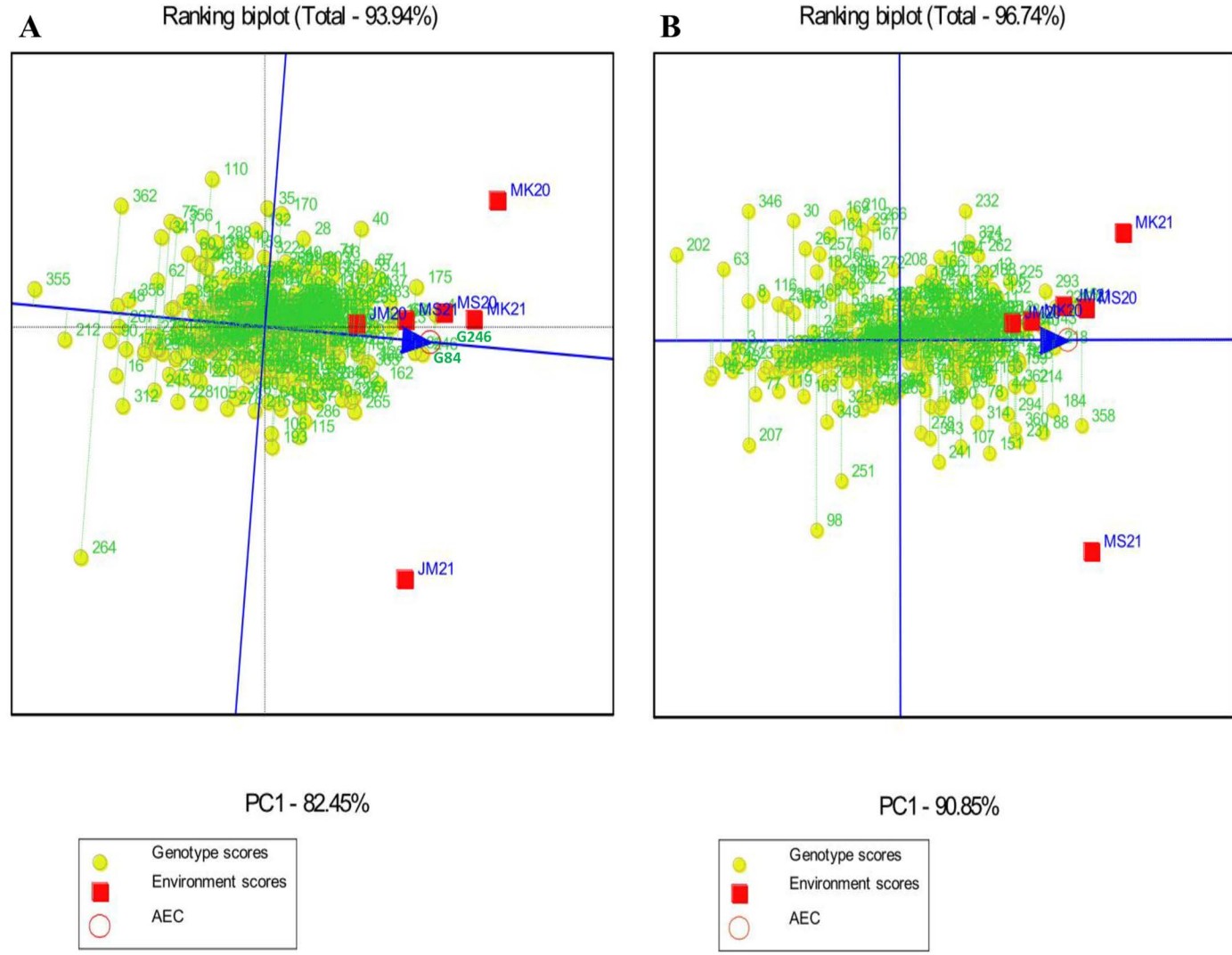

**Fig 3. Genotype main effects with genotype by environment interaction biplot genotype focus scaling displaying the stability and mean priority of the 356 sorghum landraces for (A) starch and (B) protein.** JM20 and JM21: Jimma 2020 and 2021, MK20 and MK21: Melkassa 2020 and 2021, MS20 and MS21: Miesso 2020 and 2021.

AEA vector, and G245 and G135 had the lowest mean Zn content and the longest AEA vector (Fig 4). Furthermore, genotypes identified with both high and stable starch, Fe, and Zn content, including G175, G248, G137, and G142 are deemed promising candidates for further evaluation and integration into breeding programs across diverse environments.

**Evaluation of the testing environment.** In this study, for starch content, if the target environments are distributed into mega-environments, the discriminating but non-representative test environments MK20 and JM21 can be used to select genotypes with specific adaptations. The environment of MK21 was the closest to the ideal environment and thus the best at selecting cultivars adapted to the entire region. It should be noted that more years are required to confirm that a particular test location is ideal (Fig 5A). For protein content, MS20 was the most representative and ideal test environment, as well as the closest to the AEC, while MS21 was the most discriminating environment for protein content (Fig 5B). Environments with the highest Fe content, MS21, MK20, and MK21, were representative

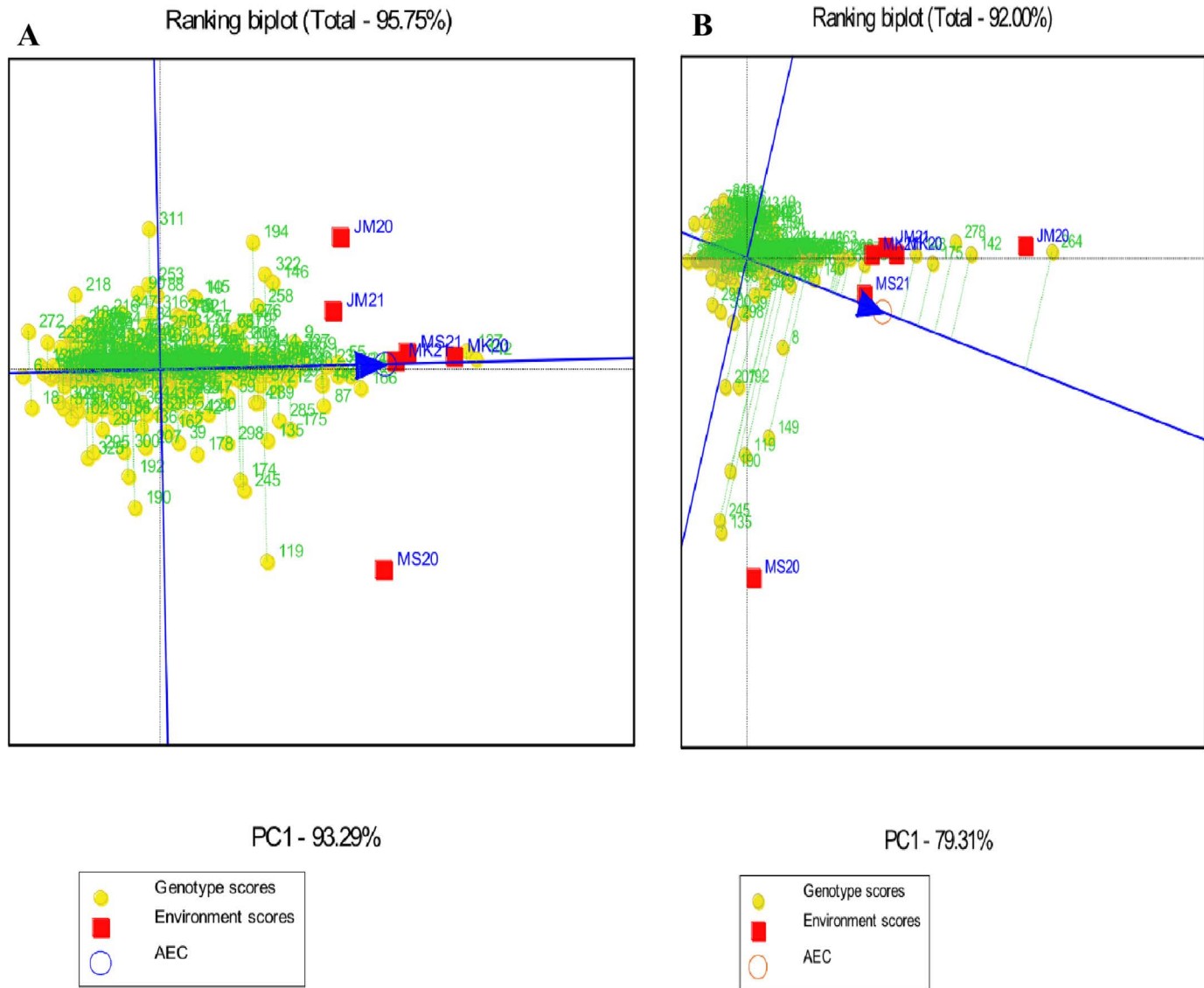

**Fig 4. Genotype main effects with genotype by environment interaction focus scaling displaying the stability and mean priority of the 356 sorghum landraces** for (A) Fe and (B) **Zn.** JM20 and JM21: Jimma 2020 and 2021, MK20 and MK21: Melkassa 2020 and 2021, MS20 and MS21: Miesso 2020 and 2021.

test environments with the smallest angles and were closest to AEC; on the other hand, MS20 and JM20 were discriminating environments for Fe concentrations (Fig 6A). The most representative environment for Zn was MS21, while the ideal and discriminating test environments were JM20 and MS20, respectively (Fig 6B).

## Association between traits

As the dataset was very large, even small correlations were significant but only correlations >0.3 are discussed here (Fig 7). Starch was positively and significantly correlated with amylose (r = 39) but negatively and significantly correlated with protein content (r = −0.39). Protein content was negatively and significantly correlated with starch (r = −0.39), Fe (r = −0.31), and Zn (r = −0.34). Fe and Zn were significantly positively correlated (r = 0.54) as well as Fe and

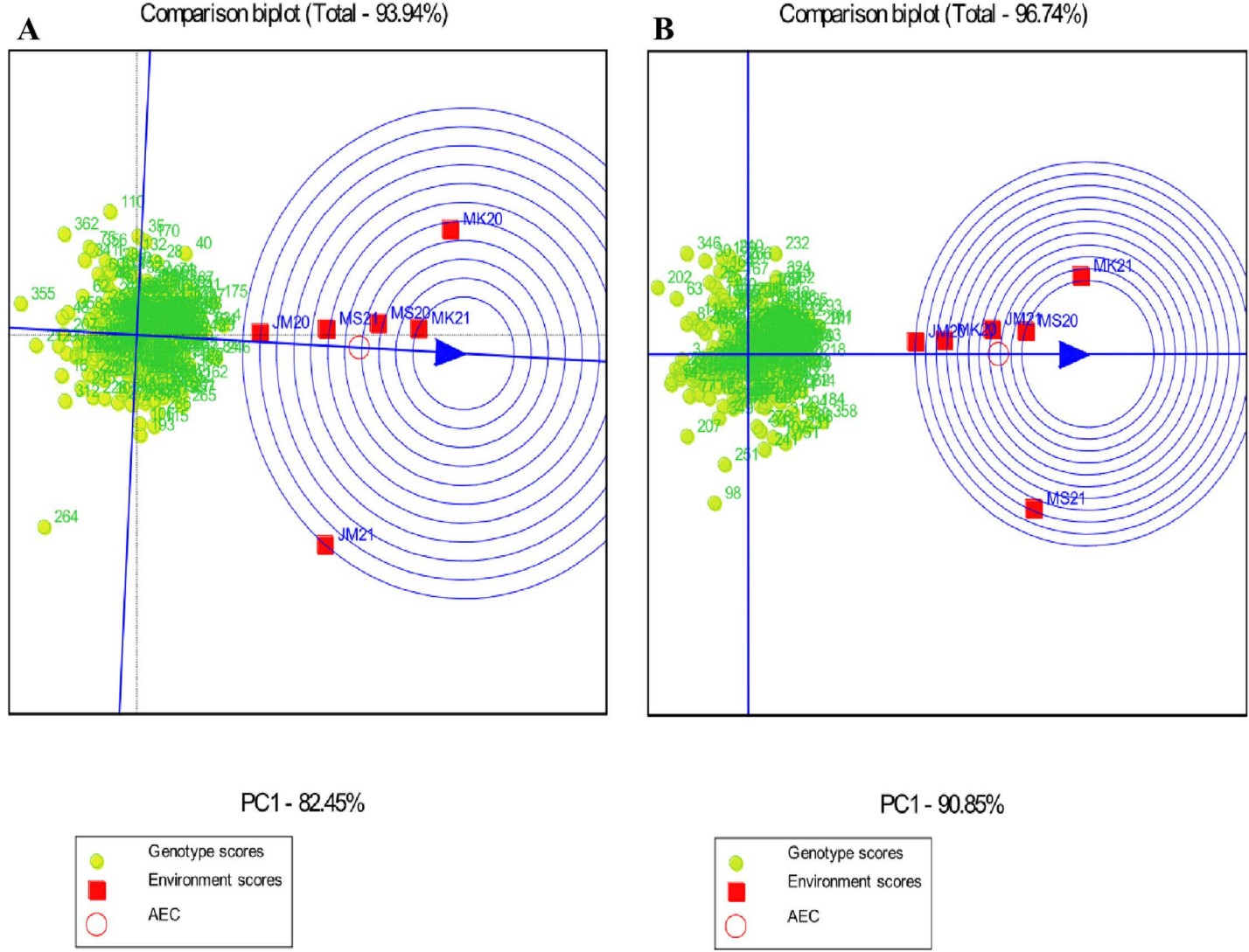

**Fig 5. The discrimination and representativeness view of the genotype main effects with genotype by environment interaction biplot used to rank test environments relative to an ideal test environment (represented by the center of the concentric circles) for (A) starch and (B) protein.** JM20 and JM21: Jimma 2020 and 2021, MK20 and MK21: Melkassa 2020 and 2021, MS20 and MS21: Miesso 2020 and 2021.

ash (r = 0.5) as were Zn and ash (r = 0.82). There was a negative and significant correlation between grain moisture content and amylose (r = −0.50).

## Discussion

The performance of promising genotypes must be repeatable and stable across years to identify and recommend superior genotypes. To accomplish this, it is crucial to conduct a MET on unbalanced data, incorporating factor analysis along with mixed models. This approach is vital for accurately gauging the relative stability and GEI of varieties. The capacity to choose the most desirable trait at a specific location is facilitated by the variation attributed to genotype and location [3]. Environmental influences on starch and amylose content have been observed in other cereal grains, such as wheat [34], and rice [35]. The amylose/amylopectin ratio holds

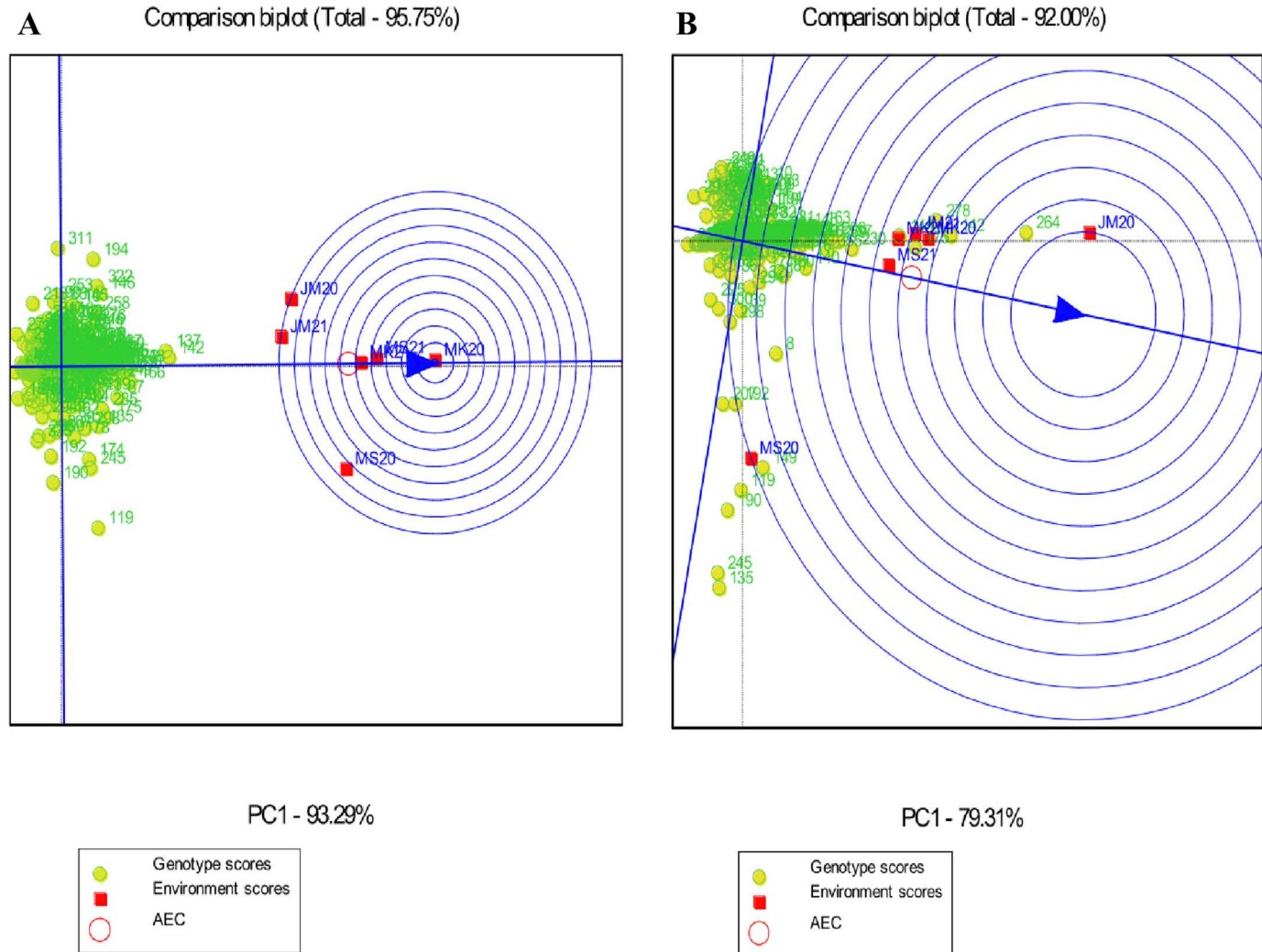

**Fig 6. The discrimination and representativeness view of the genotype main effects with genotype by environment interaction biplot used to rank test environments relative to an ideal test environment (represented by the center of the concentric circles)** for (A) Fe and (B) **Zn.** JM20 and JM21: Jimma 2020 and 2021, MK20 and MK21: Melkassa 2020 and 2021, MS20 and MS21: Miesso 2020 and 2021.

significance for various functional properties of starch, including its gelatinization profile [36]. Given the extensive genetic diversity within this sample set, the observed range in protein content was anticipated and aligned with the typical protein range found in sorghums [37].

Variation between environments was found in the comparison of GGE biplots for grain Fe and Zn content [38]. Grain Zn and Fe showed significant GEI, highlighting the need for further research to address the GEI and create sorghum landraces that are high in micronutrients. Fe and Zn concentrations in grain may vary according to soil micronutrient levels. Another reason for the high GEI for Fe and Zn content could be their heritability [39,40]. The difference in the relative performance of genotypes in different environments is a strong indicator of GEI, as well as variation in environmental conditions such as temperature, rainfall, and soil type. Starch, the main macronutrient, was reported to have a strong negative correlation with crude protein when grain macronutrients were measured on a percent dry matter basis [41]. Fe and Zn contents had a significant positive relationship over environments,

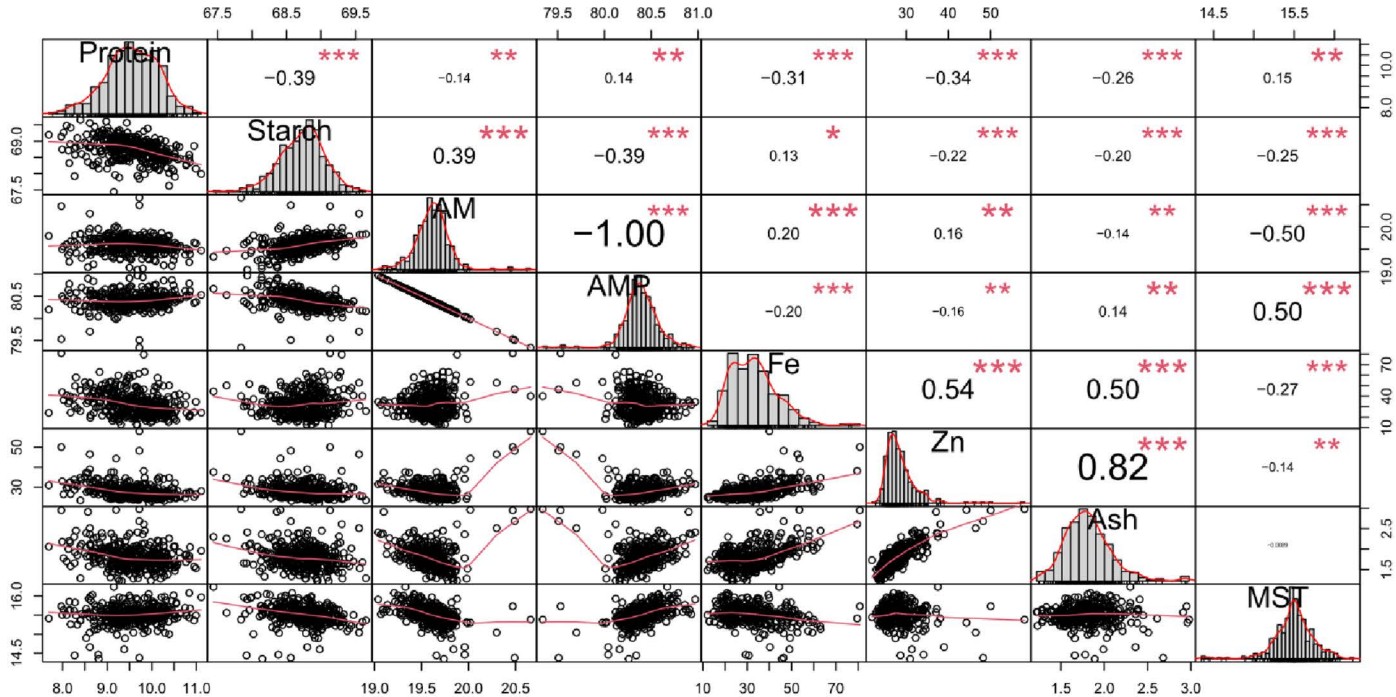

**Fig 7. The grain quality correlation coefficients of eight traits from 365 sorghum landraces.** ***: significant at 0.001 significance level, **: significant at 0.01 significance level, *: significant at 0.05 significance level, AM: amylose, AMP: amylopectin, MST: moisture, Fe: iron, Zn: zinc.

indicating the possibility of concurrently selecting for both traits [38], which was also the case in the current study.

The GGE "which-won-where" biplot was applied to distinguish top-performing genotypes by interpreting GEI, MGE clustering, and specific adaptation [9,42,43]. A polygon was drawn in this GGE biplot by connecting the vertex genotypes, which were far from the origin, with straight lines, and thus all the other genotypes were enclosed within the polygon [9,17,18]. The unstable genotypes are those that are far away from the biplot origin (vertex genotypes) and perform best or worst in different environments. Vertex genotypes, which are more responsive to environmental change, are specifically adapted genotypes that perform exceptionally well or poorly in some or all tested environments [10]. Based on the "which-won-where" biplot, the testing environments were divided into two MGEs with various high-performing genotypes for starch, protein, and Zn, but Fe only had one MGE. Environments with the highest starch content for genotypes G4, G84, and G246 in MGE1 were MK20 and MK21, MS20 and MS21, and JM20, while MGE2 had environment JM21 with G265 as the highest starch content genotype. In terms of protein, MGE1 included environments MK21, MK20, MS20, JM21, and JM20 with genotype G161 having the highest protein content, whereas MGE2 only had environment MS21 with the genotype G358 having the highest protein content.

In terms of Zn, MGE1 included environments MK21, MK20, JM21, MS21, and JM20 with the genotype with the highest concentration of Zn, G264, whereas in MGE2 environment MS20 had genotypes with the highest concentration of Zn, G245, and G135. The highest Fe-containing genotypes, G137 and G142, were combined into a single MGE for all environments. This suggests that particular genotype adaptations to MGEs occurred, successfully exploiting the GE interaction [44]. The most effective way to take advantage of the positive

GE interaction is to cluster the target environments into expressive MGEs and choose various genotypes for each MGE [18].

Various studies have explored GEI to distinguish stable and high-performing genotypes across various crops. Previous findings [45] assessed 49 sugar beet genotypes over four locations and two years, identifying stable, disease-resistant genotypes (G21, G28, and G29) and offering guidance for breeders in selecting environments and genotypes for sweet-based breeding. Similarly, [46] used AMMI and GGE biplot analyses to pinpoint stable genotypes for seed yield and mucilage traits in 30 Plantago genotypes across ten field trials, highlighting the importance of GEI in cultivar development. Genotype and environmental factors play a significant role in sugar beet traits, with $G \times E$ interactions contributing to notable yield variations [13,47]. The Palma cultivar exhibited adaptability by yielding high root and white sugar outputs under both normal and drought conditions [47]. The additive main effect and multiplicative interaction (AMMI) analysis revealed the critical role of GEI, with genotypes like F-21376 showing high stability and yield across multiple environments in sugar beet, while stability indices identified genotypes Azara and Merak as consistent yield performers [45,47].

GGE biplots are ideal for graphically assessing testing environments, as they depict the heritability of traits based on vector length and genetic correlation between environments based on the angle between vectors—two critical components of test environment assessment [17]. This model is termed GGE because it accounts for both genotype effects and genotype-environment interaction [15]. The most stable and high-ranking genotypes can be found using the GGE ranking biplot through AEC [15]. The highest starch content genotypes in the study were G246, G84, G4, and G175, while landraces with low starch contents like G264, G362, G355, and G212 experienced decreased stability due to the GEI effect. Genotypes G218, G43, and G161 scored highest in terms of protein content in this study. However, the GEI effect decreased the stability of landraces with high protein content, such as G358, G88, and G184. Although genotype G264 had the highest Zn content, the GEI effect made it less stable. The highest Fe content was found in G137 and G142. The GGE biplot model is effective for identifying top-ranking, stable genotypes across environments and selecting the best genotypes for specific mega-environments [48]. The current study found that the GGE biplot could distinguish between environments. The environments showed a similar level of discriminative power, as they were far from the biplot origin in both analyses. However, the contribution of the environments to genotype stability varied slightly.

The AEA, also known as the average-tester-axis, contains the AEC and ideal environment of all test environments and the AEA is the line that passes through the biplot origin and ideal environment. The GGE biplot displays AEA, AEC, ideal environment, and concentric circles to evaluate the tested environments. The ideal test environment should be both discriminating and representative of the target environments and be on or near the center of the concentric circles. A smaller angle with the AEA in a test environment is more representative of other test environments while test environments that are close to AEC are more representative of other environments [17,18,19]. This study highlighted MK21 as the ideal environment for starch selection, MS20 for protein, and MS21 for Fe and Zn. Discriminating environments included MK20 and JM21 for starch, MS21 for protein, and MS20 and JM20 for Fe and Zn. Extended testing is required for confirmation.

## Conclusions and recommendations

The GGE biplot analysis allows for the examination of the relative stability and performance of genotypes across environments. This analysis identified specific genotypes with both high and stable starch, protein, Fe, and Zn content. Such genotypes are valuable for breeding programs aiming to develop varieties with consistent nutritional quality across diverse conditions.

These findings have practical implications for sorghum breeding programs, which address the nutritional needs of populations across diverse environments. From assessing the adaptability of sorghum genotypes across diverse agroecological conditions, genotypes were identified with both high and stable starch, protein, Fe, and Zn content. Genotypes such as G175, G248, G137, and G142, which demonstrated superior performance in Fe, and Zn content, are regarded as excellent candidates for further evaluation and incorporation into breeding programs, offering significant potential to enhance nutritional stability across diverse agroecological regions. Similarly, G358, G218, G161, G171, and G221 exhibited the highest average protein levels. In this study, JM21 was the most discriminating test environment for starch content, while MS21 was the most discriminating for protein content.

The recommendations from this study emphasize a multifaceted approach to enhancing the adaptability and resilience of sorghum crops in diverse environments. This includes further evaluation through targeted breeding efforts and the consideration of agroecological factors.

Hence, the variability attributed to genotype and environment enables the selection of the most desirable traits in a specific environment. It is advisable to replicate this study across multiple environments to confirm and validate the obtained results.

## Supporting information

**S1 Table. Data on monthly temperatures and rainfall for two consecutive seasons (2020 and 2021).**
(DOCX)

## Acknowledgments

We are grateful for the sorghum research staff at Melkassa Agricultural Research Center.

## Author contributions

**Conceptualization:** Chalachew Endalamaw, Maryke Labuschagne.

**Data curation:** Chalachew Endalamaw, Dagmawit Tsegaye, Angeline van Biljon, Liezel Herselman, Maryke Labuschagne.

**Investigation:** Chalachew Endalamaw.

**Methodology:** Chalachew Endalamaw.

**Supervision:** Angeline van Biljon, Liezel Herselman, Maryke Labuschagne.

**Validation:** Chalachew Endalamaw, Dagmawit Tsegaye.

**Visualization:** Chalachew Endalamaw, Dagmawit Tsegaye, Liezel Herselman, Maryke Labuschagne.

**Writing – original draft:** Chalachew Endalamaw, Liezel Herselman.

**Writing – review & editing:** Chalachew Endalamaw, Dagmawit Tsegaye, Angeline van Biljon, Maryke Labuschagne.

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
