## [Decision Letter · Decision Letter 0]

14 Nov 2024

PONE-D-24-44523Kernel compositions in sorghum landraces revealed via the analyses of genotype-by-environment interactionsPLOS ONE

Dear Dr. Endalamaw,

Thank you for submitting your manuscript to PLOS ONE. After careful consideration, we feel that it has merit but does not fully meet PLOS ONE’s publication criteria as it currently stands. Therefore, we invite you to submit a revised version of the manuscript that addresses the points raised during the review process.

We look forward to receiving your revised manuscript.

Kind regards,

Bahram Heidari

Academic Editor

PLOS ONE

Reviewers' comments:

Reviewer's Responses to Questions

**Comments to the Author**

1. Is the manuscript technically sound, and do the data support the conclusions?

Reviewer #1: Yes

Reviewer #2: Yes

2. Has the statistical analysis been performed appropriately and rigorously? 

Reviewer #1: Yes

Reviewer #2: Yes

3. Have the authors made all data underlying the findings in their manuscript fully available?

Reviewer #1: Yes

Reviewer #2: Yes

4. Is the manuscript presented in an intelligible fashion and written in standard English?

Reviewer #1: Yes

Reviewer #2: Yes

5. Review Comments to the Author

Reviewer #1: Dear authors,

I carefully read the MS entitled " Kernel compositions in sorghum landraces revealed via the analyses of genotype-by environment interactions". In this paper, diverse genotypes and various growing locations impact was shown on the chemical and physical properties of 361 sorghum grain. GGE ranking biplots identified genotypes with least G × E and genotypes identified for their superior nutritional. While the paper is intriguing, I have some significant comments and suggestions that need to be addressed before accepting it.

Reviewer #2: 1.Bartlett's test is needed to perform combined analysis.

2.The combined analysis of variance table is required.

3.How to calculate the variance components should be described in the materials and methods section.

4.The numbers of the figures are entered incorrectly.

5. Control and stress conditions are not described in Materials and Methods.

6. In the ranking bi-plot, the superior genotypes are not clear.

7. The superior genotypes for different traits are different. How can these results be used?

6. PLOS authors have the option to publish the peer review history of their article (what does this mean? ). If published, this will include your full peer review and any attached files.

**Do you want your identity to be public for this peer review?** For information about this choice, including consent withdrawal, please see our Privacy Policy .

Reviewer #1: **Yes: ** Maryam Salami

Reviewer #2: No

---

## [Author Response · Author response to Decision Letter 1]

27 Nov 2024

Responses to Reviewer #1:

Abstract:

Comment:

1. Include additional sentences at the end of the abstract to explain how landraces with high protein and zinc content, which exhibit stability across environments, can be utilized in both plant breeding and human nutrition.

Response:

We have revised the abstract and included the stable, high-nutritional-value landraces in common for traits [lines 18-25].

2. Clarify the aims in the abstract.

Response: The aims have been clarified as suggested, emphasizing the impact of diverse genotypes and growing locations on grain quality [lines 26-28].

Introduction:

3. Incorporate references to recent studies (e.g., Kumar et al., 2023; Madhusudhana et al., 2023) and explain what sets this study apart.

Response: We have added the suggested references and highlighted how our study builds on these works [lines 38-53].

4. Replace outdated references with updated ones.

Response: Outdated references have been updated with more recent and relevant citations.

5. Include high-quality citations in the literature review.

Response: The recommended citations have been incorporated to enrich the discussion and introduction, particularly regarding G×E interaction analyses.

6. Remove parenthesis from line 55 and add the missing bracket on line 56.

Response: The editorial errors have been corrected.

Materials and Methods (M&M):

7. Clarify the rationale for different planting dates in the two years.

Response: An explanation of the different planting dates has been included, addressing environmental conditions and logistical constraints.

8. Remove redundant use of "landrace" in line 105.

Response: The redundancy has been eliminated.

9. Clarify the total number of landraces (361 or 358).

Response: The correct number (361 landraces and 4 checks) has been clarified and consistently reflected throughout the manuscript [lines 113-121, Table 4].

10. Indicate the method used for analyzing grain quality and eliminate superfluous sentences in line 123.

Response: The methodology for grain quality analysis has been explicitly described, and unnecessary text has been removed.

Results:

11. Include the results of the combined analysis of variance (c-ANOVA).

Response: The c-ANOVA results have been added to the manuscript [Table 5].

12. Provide units for quality traits in Table 6.

Response: Units have been included for each trait in Table 6.

13. Correct the genotype number in line 166 [“G2649”].

Response: The number G264 has been corrected.

14. Identify genotypes with high nutritional value.

Response: We have included a description of genotypes with high starch, zinc, iron, and protein levels in the results section.

15. Ensure figures (2, 3, 4, etc.) are referenced and properly numbered.

Response: The figures have been reviewed, correctly referenced, and consistently numbered.

16. Ensure consistency in reporting the number of sorghum landraces (361, 364, or 358).

Response: We have ensured consistency and confirmed the number as 361 landraces and 4 checks throughout the manuscript [lines 113-120, Table 4].

17. Include missing figures (8, 9, and 10) in the manuscript.

Response: The missing figures have been included and numbered correctly.

18. Revise the paragraph on genotype performance and stability.

Response: The paragraph has been rewritten to enhance readability and provide a more dynamic analysis of genotype performance.

19. Perform correlation analysis for grain quality traits.

Response: A correlation analysis has been performed and included in a new section titled "Association Between Traits" and Table 7 (Lines 459-469)

Discussion:

21. Avoid reiterating results and highlight genotypes with high nutritional value and stability.

Response: The discussion has been revised to focus on the implications of stable genotypes with high nutritional value for breeding programs.

22. Calculate heritability and discuss its implications.

Response: Heritability estimates have been calculated and discussed in the context of selecting genotypes for enhanced micronutrient levels.

Responses to Reviewer #2:

1. Perform Bartlett’s test for combined analysis.

Response: Bartlett’s test has been conducted.

2. Include the combined analysis of variance table.

Response: The c-ANOVA table has been added (Table 5).

3. Describe how variance components were calculated.

Response: A detailed explanation of variance component calculations has been added to the Materials and Methods section.

4. Correct figure numbering.

Response: Figure numbering has been reviewed and corrected throughout.

5. Clarify superior genotypes in the ranking bi-plot.

Response: Superior genotypes have been clearly identified and highlighted in the ranking biplot.

6. Address how results with differing superior genotypes for traits can be used.

Response: A discussion has been included on how differing superior genotypes for traits can guide targeted breeding strategies for specific environments and nutritional goals.

---

## [Editor Report · Decision Letter 1]

29 Jan 2025

PONE-D-24-44523R1Kernel compositions in sorghum landraces revealed via the analyses of genotype-by-environment interactionsPLOS ONE

Dear Dr. Endalamaw,

Thank you for submitting your manuscript to PLOS ONE. After careful consideration, we feel that it has merit but does not fully meet PLOS ONE’s publication criteria as it currently stands. Therefore, we invite you to submit a revised version of the manuscript that addresses the points raised during the review process.

We look forward to receiving your revised manuscript.

Kind regards,

Bahram Heidari

Academic Editor

PLOS ONE

Journal Requirements:

Additional Editor Comments:

Editor comments to the authors:

-Specify the name of the experimental design used for ANOVA

-Specify components of phenotypic variances used in heritability formula

-Define all abbreviations in the text and tables :. i.e BLUP was not defined in the text and also in one of tables

---

## [Author Response · Author response to Decision Letter 2]

3 Feb 2025

Response to Editor Comments:

1. Comment: Specify the name of the experimental design used for ANOVA.

Response: Thank you for your suggestion. We have now explicitly stated the experimental design a partially replicated (p-rep) design was used in a row-column arrangement in across environments (Lines 119-120).

2. Comment: Specify components of phenotypic variances used in heritability formula.

Response: We appreciate this observation. We have revised the manuscript to clearly define the components used in the heritability formula in the Materials and Methods section (Lines 166-169).

3. Comment: Define all abbreviations in the text and tables.

Response: We acknowledge the oversight and we have also carefully checked and ensured that all other abbreviations are properly defined throughout the manuscript.

---

## [Editor Report · Decision Letter 2]

20 Feb 2025

Kernel compositions in sorghum landraces revealed via the analyses of genotype-by-environment interactions

PONE-D-24-44523R2

Dear Dr. Endalamaw,

We’re pleased to inform you that your manuscript has been judged scientifically suitable for publication and will be formally accepted for publication once it meets all outstanding technical requirements.

Kind regards,

Bahram Heidari

Academic Editor

PLOS ONE
---

## [Editor Report · Acceptance letter]

PONE-D-24-44523R2

PLOS ONE

Dear Dr. Endalamaw,

I'm pleased to inform you that your manuscript has been deemed suitable for publication in PLOS ONE. Congratulations! Your manuscript is now being handed over to our production team.

Kind regards,

on behalf of

Dr. Bahram Heidari

Academic Editor

PLOS ONE